# Numerical Analysis of Viscous Dissipation in Microchannel Sensor Based on Phononic Crystal

**DOI:** 10.3390/mi12080994

**Published:** 2021-08-21

**Authors:** Juxing He, Honglang Li, Yahui Tian, Qiaozhen Zhang, Zixiao Lu, Jianyu Lan

**Affiliations:** 1National Center for Nanoscience and Technology, Beijing 100190, China; hejuxing20@mails.ucas.ac.cn (J.H.); luzx2019@nanoctr.cn (Z.L.); 2University of Chinese Academy of Sciences, Beijing 100190, China; 3Center for Excellence in Nano Sciences, Chinese Academy of Sciences, Beijing 101400, China; 4Institute of Acoustics, Chinese Academy of Sciences, Beijing 100190, China; 5School of Information, Mechanical and Electrical Engineering, Shanghai Normal University, Shanghai 200234, China; zhangqz@shnu.edu.cn; 6State Key Laboratory of Space Power-Source Technology, Shanghai Institute of Space Power-Sources, Shanghai 200245, China; jianyu_lan@163.com

**Keywords:** phononic crystal, microchannel, viscous dissipation, chemical sensor

## Abstract

Phononic crystals with phononic band gaps varying in different parameters represent a promising structure for sensing. Equipping microchannel sensors with phononic crystals has also become a great area of interest in research. For building a microchannels system compatible with conventional micro-electro-mechanical system (MEMS) technology, SU-8 is an optimal choice, because it has been used in both fields for a long time. However, its mechanical properties are greatly affected by temperature, as this affects the phononic bands of the phononic crystal. With this in mind, the viscous dissipation in microchannels of flowing liquid is required for application. To solve the problem of viscous dissipation, this article proposes a simulation model that considers the heat transfer between fluid and microchannel and analyzes the frequency domain properties of phononic crystals. The results show that when the channel length reaches 1 mm, the frequency shift caused by viscous dissipation will significantly affect detecting accuracy. Furthermore, the temperature gradient also introduces some weak passbands into the band gap. This article proves that viscous dissipation does influence the band gap of phononic crystal chemical sensors and highlights the necessity of temperature compensation in calibration. This work may promote the application of microchannel chemical sensors in the future.

## 1. Introduction

Phononic crystals (PnC) are analogous to photonic crystals, which are composed of periodic arrays of elastic materials. In phononic crystals, phononic bandgaps can be generated throughout the Brillouin zone, which make it possible to realize a vibrationless environment at a specific frequency. Given diverse parameters influencing the band’s structure, it also has applications in sensing [1,2,3]. As soon as the concept of phononic crystals was proposed, it began to attract a lot of attention. The related theoretical computation methodology advanced rapidly, and the structure was changed from the initial tetragonal superlattice [4] formed by cylindrical inclusions to more complex structures [5].

In binary cermet topology, phononic bandgaps can be generated by materials with high sound velocity as the host and material with low sound velocity as the internal periodic array [6]. Such properties suggest that these bandgaps can be used to measure fluid properties in microchannel systems, and these properties can also be chemical related. A step further, chemical sensors can be realized following the post-processing of data. However, producing a microchannel resonant structure compatible with conventional MEMS processes is also a problem [7]. SU-8 can help to resolve this, because it is a common building material used in microchannels and is also used as a photoresistor in traditional MEMS processes.

In the field of microchannel chemical sensors, the previous works are mainly based on the cavity mode [5,8,9], in which the liquid flow channel does not go through the phononic crystal. The liquid in cavities can be seen as static, so little attention is paid to the thermal effect. Researchers have focused more on the progression of band design and validation of the system’s theory. The use of cavity mode means that repeated liquid injection is needed throughout continuous measurement, and the injection liquid is not guaranteed to fill the whole cavity. Furthermore, it is difficult to determine whether the previously measured solution has an impact on the latest one. Therefore, it may be necessary to allow the liquid to flow directly through the phononic crystal. The chosen applications require rapid and accurate analyses of solutions, and so the liquid phase in the microchannel sensor should be mobile, which will certainly introduce microfluid-related phenomena. Moreover, in a microchannel made of SU-8, the thermal effect is highlighted. Because the mechanical properties of SU-8 change significantly with temperature, this will also change the phononic bands [10]. Against this background, consideration of the thermal field is necessary, which has also been a research focus for a long time [11,12,13]. In regard to microchannels, there is one factor that is normally ignored in macroscopic channels but is magnified as the channel size decreases—viscous dissipation. It is derived from the velocity gradient around the microchannel wall in pressure-driven flow, which is negligible in macroscopic flows with a low Reynolds number [14]. To solve the problem of viscous dissipation, this study proposes an analysis model considering the heat transfer between fluid and microchannel and analyzes the frequency domain properties of phononic crystals at the same time.

## 2. Simulating Model

The eigen frequency of a finite periodic structure is dependent on the frequency band with a pure imaginary propagation coefficient, which means its passband is only related to frequency. COMSOL multiphysics with the finite element method (FEM) can be used to analyze the structure’s waveguide. The advantages of this lie in its flexibility. After providing appropriate element routines, various structural configurations can be examined [15]. The subjects of this simulation study are SU-8 and water, in which the viscosity and heat loss caused by the acoustic field are relatively small. Therefore, the governing equation of acoustics is shown in Equation (1).
(1)$$\nabla \cdot \left[-\frac{1}{\rho_{0}}\left(\nabla p-q\right)\right]-\frac{\omega^{2}p}{\rho_{0}c^{2}}=Q.$$

A crossover schematic diagram of components used in the simulation is shown in Figure 1 [8]. The channel length determines the heat generated by viscous dissipation, so 3D channels of 100 μm and 1 mm length are simulated. The channels are made of SU-8, which is compatible with traditional MEMS processes. The separation of the piezoelectric crystal from the detecting channel precludes the scattering of acoustic waves into the piezoelectric crystal’s bulk and resolves the difficulty of machining the microchannel on a piezoelectric crystal.

As the focus of this simulation is the effect of temperature on phononic crystal microchannels, an acoustic wave-generating device composed of an interdigital transducer and piezoelectric crystal is replaced with a background pressure field at blk3 as the sound pressure source, with the absolute pressure set as $10^{5}$ Pa. The structure of the component is shown in Figure 2. The acoustic wave will propagate parallel to the x axis, while the fluid will flow parallel to the y axis. Ext1 and ext2 are set as perfectly matching layers (PML) to absorb the energy of reflective sound, which facilitates the simulation. In reference [8], micropipes are used to inject liquid into the cavities, and the liquid in them is regarded as static. The paper also mentioned that cavities may not be filled completely with liquid. In practical applications, due to the low flow rate of liquid in the microchannel, the amount to be detected is very low. Repeated injections for each detection may amplify this shortage, so the detection of flowing liquid is a problem that must be considered in order to improve detection speed. In this situation, it may be beneficial to allow the fluid to flow through the phononic crystal. Therefore, this study simulates the viscous dissipation effect of a microchannel chemical sensor based on the structure of phononic crystal.

Arr1 is set as the fluid domain, and the property we mainly focus on is sound velocity. As for the temperature, dynamic viscosity coefficient and other parameters, we adopt the corresponding values of water that are built into the software. The other domains are set as SU-8, and its parameters are set as shown in Table 1. The physical fields used are pressure acoustics, frequency domain and conjugate heat transfer. The fluid in the field is set as an incompressible laminar flow, with viscous dissipation. The inlet is set to be driven by a static pressure of $10^{5}$ Pa, with suppressed backflow. The outlet is set as a fully developed flow whose average velocity is 1 m/s. The mesh structure for all domains except PML is normal free tetrahedral, celebrated for its general physical qualities, while the fluid domain’s structure is celebrated for its fluid dynamics. The element size of the body in the fluid domains is coarse, while it is fine in the boundaries. Corner refinement and boundary layers are also applied in fluid domains. Lastly, the PML comprises a distribution of eight elements for sweeping.

## 3. Simulating Result 

Before viscous dissipation, the relevant simulation is performed, and the frequency domain response of the model to liquids with different sound velocity properties (the sound velocity in an aqueous solution of organic matter varies with the ratio of solvents) is tested in a 100 μm channel. In the pressure acoustic frequency domain study, the absolute sound pressure on the upper interface of ext2 is integrated, which illustrates the change in the passband in the phononic crystal. Before that, a band diagram of the tetragonal superlattice element is constructed as a 2D model is measured, as shown in Figure 3. For propagation, the acoustic wave needs not only an x axial component but also a z axial component in order to travel from blk3 into the phononic crystal. Therefore, the phononic bandgap should be located at the frequency of 12–14 MHz. This result is also consistent with Equations (2)–(4), used for band calculation, as shown below [4].
(2)$$\delta \rho =\frac{\Delta \rho }{\bar{\rho }}=\frac{\rho a/\rho b-1}{f\rho a/\rho b+1-f},$$
(3)$$\delta \tau =\frac{\Delta \tau }{\bar{\tau }}=\frac{\rho_{a}c_{la}^{2}/\rho_{b}c_{lb}^{2}-1}{f\rho_{a}c_{la}^{2}/\rho_{b}c_{lb}^{2}+1-f},$$
(4)$$\left({\left|k+g\right|}^{2}-\Omega^{2}\right)u_{k}\left(g\right)+\underset{g^{\prime }\neq g}{\sum }F\left(g-g^{\prime }\right)\left[\left(\delta \tau \right)\left(k+g\right)\cdot \left(k+g^{\prime }\right)-\left(\delta \rho \right)\Omega^{2}\right]\times u_{k}\left(g^{\prime }\right)=0.$$

The equations above are used to calculate the eigenvalue Ω(k) and eigenvector $u_{k}\left(g\right)$ of the structure. k is the wave vector; g is the inverted lattice vector of the superstructure; $\rho_{a}$ and $\rho_{b}$ are the densities of the host material and the inclusion material, respectively; $c_{la}$ and $c_{lb}$ are the wave velocities of the longitudinal wave in the host material and the inclusion material, respectively; and f is the filling ratio of the inclusion in the host material.

The analysis of frequency domain is then performed. The result is shown in Figure 4, where the bandgap is essentially consistent with the result in Figure 3. It can also be seen that the change in sound velocity of the working fluid in the channel does affect the phononic band, and the magnitude of frequency shift is consistent with the simulation in reference [8]. The local resonance peak intensities are different in Figure 4 due to the different step sizes chosen in the two processes, which may lead to absences in some data points.

According to reference [10], the mechanical properties of SU-8 change significantly with temperature, which means that the sound velocity also changes with temperature. Therefore, the change in temperature will also affect the band structure of the phononic crystal. As shown in Figure 5, the edge of the band changes significantly with temperature. This also explains the need for a study of viscous thermal effect. From the analysis of the data, the frequency shift caused by the change in sound velocity from 1500 m/s to 1490 m/s is 0.02 MHz, while the shift is 0.05 MHz for temperature changes from 293.15 K to 294.15 K. Therefore, if the liquid channel passes directly through the phononic crystal, the thermal effect must be taken into consideration. It is worth noting that the local resonance peak will contact the right edge of the gap if the temperature rises to 355 K, which indicates that our system may no longer be able to perform as a chemical sensor even with proper temperature compensation measures. Additionally, the thickness of the side wall also influences the location of the local resonance peak.
(5)$$\rho C_{p}U_{ave}\frac{dT_{m}}{dx}=k\cdot \frac{d^{2}T_{m}}{dx^{2}}+\frac{\mu \iint_{A}\Phi dA}{A}$$

Above are fluid heat transfer equations for a rectangular microchannel [17], where ρ is the fluid density, $C_{p}$ is the liquid isobaric heat capacity, $U_{ave}$ is the average velocity for a cross-section of the channel, $T_{m}$ is the mean temperature of a cross-section of the channel, k is the thermal conductivity, μ is the fluid viscosity, Φ is the viscous dissipation function, and A is the cross-sectional area of the channel.

The viscous dissipation effect of fluids is often neglected in macroscopic channels. Equation (5) illustrates that the viscous dissipation value is inversely proportional to the cross-sectional area of the channel for a given fluid. The viscous dissipation increases as the channel becomes smaller. Therefore, in the microchannel, even fluids with a laminar Reynolds number can cause considerable temperature field variations [17]. After this, the viscous heat of the fluid in the flow channel and the conjugate heat transfer between the fluid and the channel are simulated. The fluid is set as an incompressible flow, the upstream temperature of which is 293.15 K. The heat flux on the outside wall of the microchannel is set as natural heat convection, and the bottom face temperatures of ext1 and ext2 are set as 293.15 K, because the piezoelectric substrate is estimated to be half-infinite. The remaining walls are thermally isolated, and the steady state is studied. The result for a 100 μm channel is shown in Figure 6. In Figure 6a, along the flow direction of the liquid, the temperature gradually rises with the viscous force. The velocity gradient in the pressure-driven flow is steeper near the channel wall, meaning the SU-8 wall between the channels at the exit reaches the highest temperature, as shown in Figure 6b. As the channel is relatively short, the temperature change is of the order $10^{-3}$ K. The magnitude of the temperature here agrees with the result in reference [17].

In the previous simulation, the temperature change is not obvious, so the length of the channel is extended to 1 mm. According to reference [18], an increase in thickness may lead to the closure of the bandgap, which corresponds to the increase in length of the microchannel in our system. Therefore, it is necessary to analyze the response in the full frequency domain after increasing the length of the channel. As shown in Figure 7, the frequency domain response and the phononic bandgap position remain essentially unchanged, indicating that the structure can still form a phononic bandgap near the 13 MHz frequency and can act as a chemical sensor when its length is extended to 1 mm.

Conjugate heat transfer in the 1 mm microchannel is simulated. As shown in Figure 8, the temperature rises to 0.01 K, which verifies the temperature increase with the increasing length of the channel [17]. When the length of the channel increases to 1 mm, the temperature change pattern is essentially the same as that of the 100 μm channel. The difference is that the temperature of the liquid in the runner also increases somewhat, making it significantly higher than the inflow temperature. This reason is that the longer channel weakens the role of inflow temperature and relatively high flowing velocity. Moreover, the distinct temperature drop at the side walls is caused by the consistent temperature boundaries below.

Because the frequency spectrum of the 1-mm-long channel is essentially the same as that of the 100 μm channel, only the phononic band edge can be simulated accurately. As shown in Figure 9b, the frequency shift of the phononic band edge in the 1 mm channel reaches about 0.01 MHz, which is about 50% of the frequency shift for the sound velocity changing from 1500 m/s to 1490 m/s, indicating that viscous dissipation is a nonnegligible factor in the calibration of accurate chemical measurement. The sound velocity accuracy of the sensor should be 5 m/s without temperature compensation in our system. This can be improved by slowing the flow velocity, as this will reduce the viscous fluid’s heat. In addition, it is worth noting that the temperature gradient in the fluid and the SU-8 adhesive caused by the viscous dissipation effect may weaken the shielding effect of the phononic crystal against the acoustic wave in the bandgap, resulting in a weak pass band inside the bandgap (Figure 9a). This may be due to the inhomogeneity in mechanical properties along the flow orientation resulting from the temperature gradient.

## 4. Conclusions

A microchannel chemical sensor based on a phononic crystal has excellent application prospect. The main principle is that the band edge structure of the phononic crystal changes with the change in sound velocity caused by different chemical compositions in the liquid. A phononic crystal chemical sensor built in cavity mode may face some problems, such as mutual interference in multiple measurements, difficulty in multiple sample injection, and difficulty in filling the cavity repeatedly. In order to meet the practical needs, it may be necessary to drive the liquid flow through the phononic crystal by means of pressure or electroosmosis. In the microchannel, the viscous dissipation of flow with a low Reynolds number can also cause obvious temperature field changes, which will have a significant impact on the mechanical properties of SU-8. Therefore, this article proposes a model for the viscous dissipation and the conjugate heat transfer between fluid and microchannel. Furthermore, the frequency domain properties of phononic crystal are also analyzed. Through numerical analysis, it can be found that when the channel length reaches 1 mm, the frequency shift brought about by viscous dissipation will significantly affect the detection accuracy. Therefore, in standard band edge calibration, the standard operating conditions of the sensor, such as the Reynolds number and the temperature, should be clearly defined. When making high-precision measurements, temperature compensation mechanisms should be considered. The temperature gradient related to the viscous dissipation effect may give rise to a weak passband in the phononic gap, whose effect may be more obvious in a longer flow channel. This work has proposed a reliable model and proven that viscous dissipation does influence the band gap of phononic crystal chemical sensors. Moreover, it declares the necessity of temperature compensation during calibration, which will promote the application of microchannel chemical sensors in the future.

## Figures and Tables

**Figure 1 micromachines-12-00994-f001:**
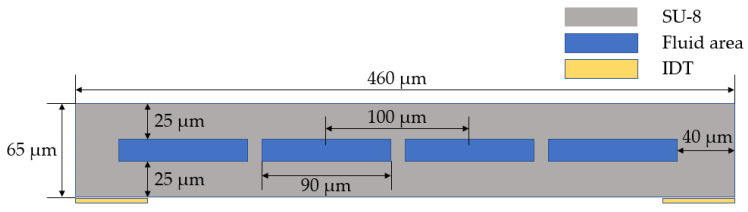
Crossover schematic diagram of the channel [8].

**Figure 2 micromachines-12-00994-f002:**
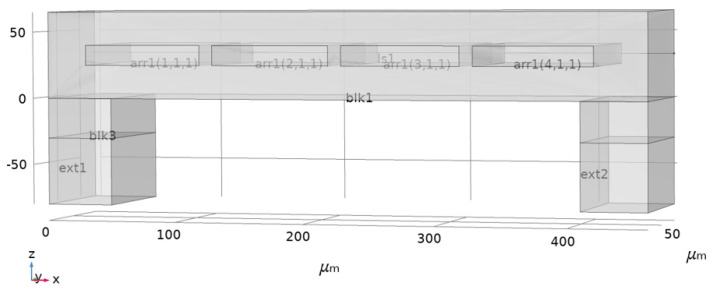
Schematic diagram of the model.

**Figure 3 micromachines-12-00994-f003:**
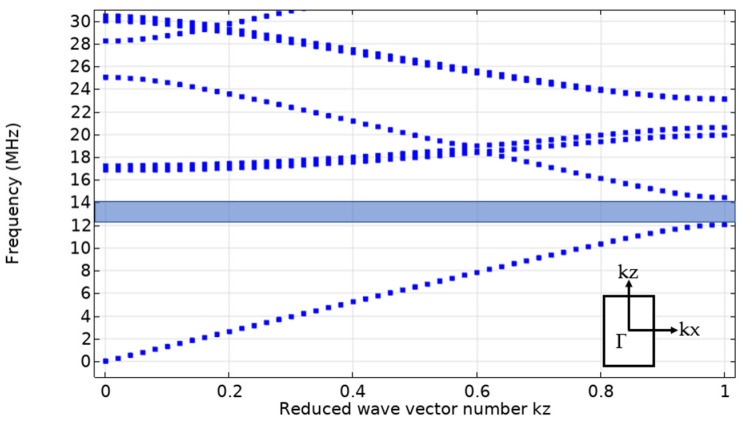
Band diagram of the phononic crystal periodic element (along kz).

**Figure 4 micromachines-12-00994-f004:**
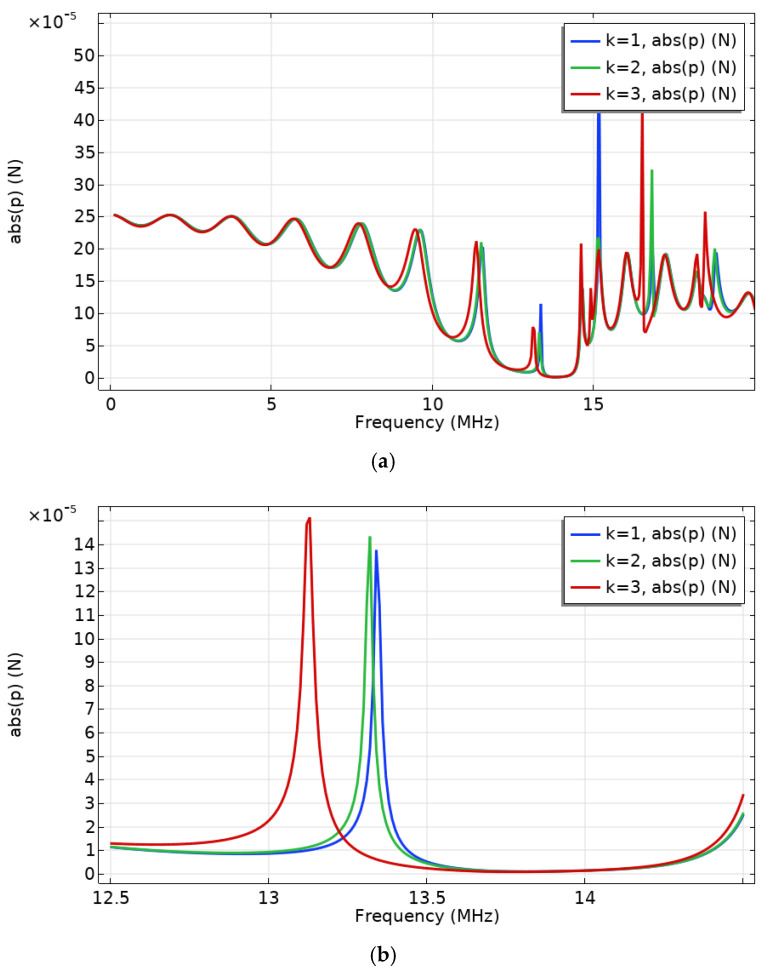
Frequency domain response at 293.15 K (k = 1, 2, 3 correspond to the liquid sound velocity = 1500, 1490, 1420 m/s, which may correspond to the sound velocity values of aqueous solutions of different concentrations of a specific solvent/different solvents of a specific concentration). (**a**) Response for the range 0.1−25 MHz. (**b**) Response for the range 12.5−14.5 MHz (where the edge of the phononic bandgap is located).

**Figure 5 micromachines-12-00994-f005:**
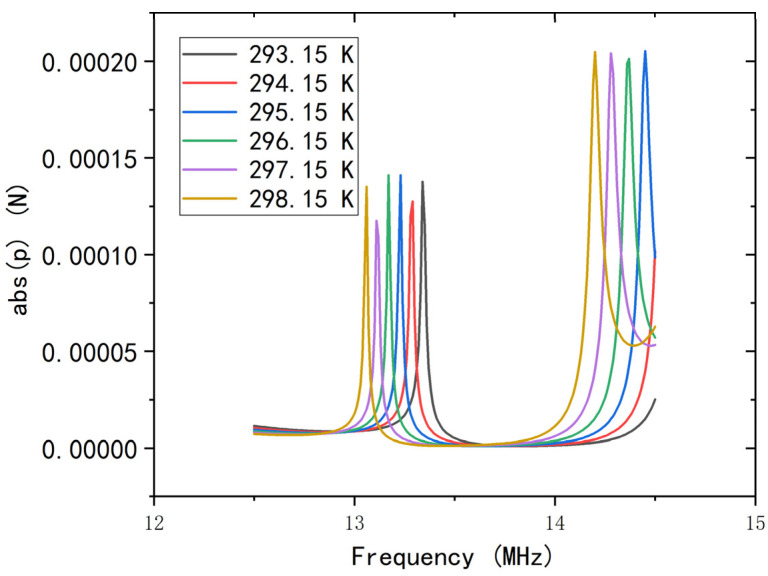
The change in the edge of the bandgap with changes in temperature, with a liquid sound velocity of 1500 m/s.

**Figure 6 micromachines-12-00994-f006:**
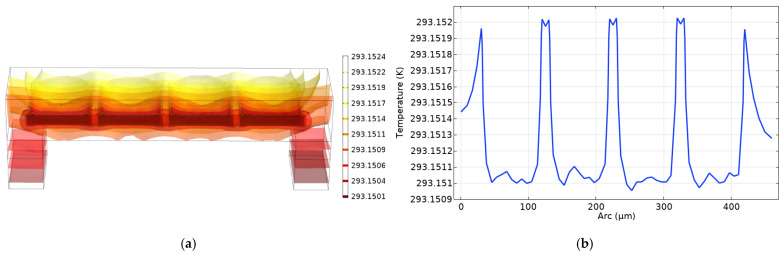
(**a**) Isothermal surface of a 100 μm channel. (**b**) Temperature curve along the horizontal midline of the 100 μm channel’s exit.

**Figure 7 micromachines-12-00994-f007:**
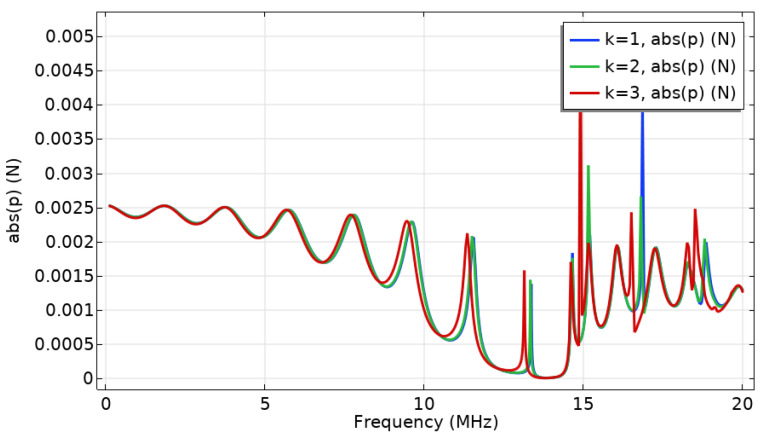
Rough frequency domain analysis of a 1 mm channel at 293.15 K (k = 1, 2, 3 correspond to the liquid sound velocity = 1500, 1490, 1420 m/s).

**Figure 8 micromachines-12-00994-f008:**
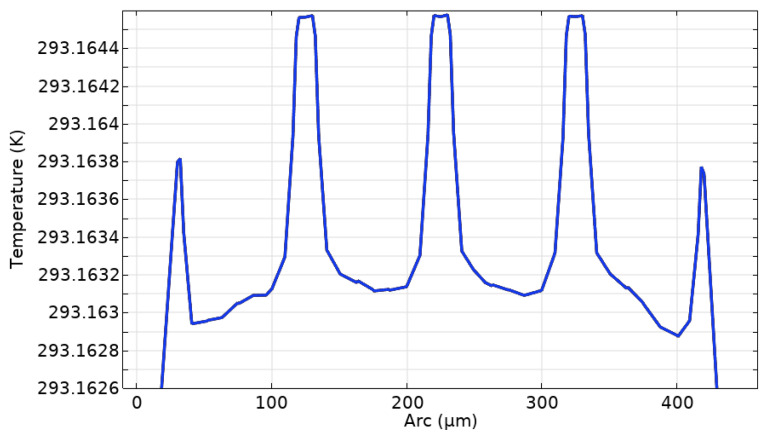
Temperature curve along the horizontal midline of the 1 mm channel’s exit.

**Figure 9 micromachines-12-00994-f009:**
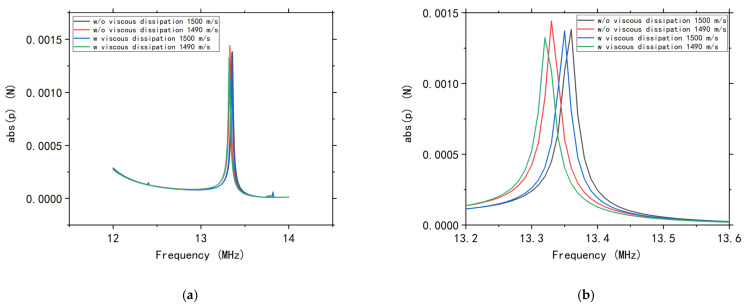
Comparison of the frequency shift caused by sound velocity change and viscous dissipation. (**a**) Response in the range of 12−14 MHz. (**b**) Response for the range of 13.2−13.6 MHz.

**Table 1 micromachines-12-00994-t001:** Physical parameters of SU-8 (Young’s modulus and Poisson’s ratio are cubic polynomially fitted using the data in reference [10] (the SU-8 in this reference is not post-baked, so the Young’s modulus is relatively low)).

Physical Parameter	Expression	Reference
Young’s modulus	−4.087e-7*T^3+6.182e-4*T^2-0.3115*T+52.411 GPa	[10]
Poisson’s ratio	−3.82e-8*T^3+4.32e-5*T^2-0.0152*T+2.05	[10]
Coefficient of thermal expansion	52 ppm/K	[16]
Specific heat	1500 J/(kg *K)	[14]
Thermal conductivity	0.2 W/(m *K)	[14]

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
