# Peer review of "Numerical Analysis of Viscous Dissipation in Microchannel Sensor Based on Phononic Crystal"

_micromachines, 2021, doi:10.3390/mi12080994_

Round 1

Reviewer 1 Report

In the paper „Simulating Study of Viscous Dissipation in Microchannel of Chemical Sensor Based on Phononic Crystal” the authors address the interesting topic, if viscous dissipation in microchannels does influence the reading of phononic crystal.

I would highly recommend using SI units and not units like atm.

The geometry is inappropriately described. There is no hint, if the simulation is 2D or 3D. There is also no indication which kind of grid was used and what its resolution is.

I found no indication how liquid flow was simulated. Also, the average flow velocity of 1m/s seems to be unrealistic high. My estimate is that there will be a pressure drop of about 600bar on the 1mm. Taking into account that there will be supply channels needed the chances are high that one will not find any pump providing this flow. With such high pressures also a considerable deformation of the 25um thick walls can be expected which will massively impact on the acoustics.

To take a circular microchannel approximation in equation 9 for approximating a rectangular cross section with aspect ratio is inappropriate.

Also not all thermal boundary conditions are fully explained.

There will also be quite some heat transfer through the polymer material. To only use natural heat convection for the outer walls is insufficient. Already work at the end of the 1990s showed that micro structured heat exchangers follow a different logic than macroscopic ones, e.g. their optimal building material would be ceramics or polymers.

If the temperature change of a 100um channel is found to be 10^-4K a 1mm channel should lead to an 1mK increase in first approximation. I highly doubt that any experimental system can be kept to stable temperature by more than 1mK having those high fluid flow rates and pressure drops.

The temperature influence of the piezoelectrics is also fully ignored and in practice has a high chance of having a higher impact than the effects investigated here.

Adding all this up I do not see that this is a reasonable concept. If the authors can show experimental results fitting to the simulation of this paper the paper might be reconsidered.

But even in this case the title must be changed. The paper does not present a chemical sensor. There is no selective element that allows the differentiation of different chemicals. In the best case it might be a viscosimeter or thermometer. But, both measurements are physical meter and no chemical sensor.

There is a long list of minor details which would have to be changed, if the research setting would have made sense, i.e. quality of some figures, text of figure captions, unit systems (sometimes K sometimes °C), …

Author Response

Thank you so much for your suggestion about revising. We have revised our article carefully according to your suggestion. Please see the attachment for point-by-point response.

Reviewer 2 Report

To improve the quality of the manuscript, the following comments should be addressed by the authors.

  • There are some serious grammatical and spelling mistakes in the manuscript. The author should proofread the manuscript from a more fluent English speaker.
  • The word “Simulating” in the manuscript title looks awkward therefore I would suggest revising the title.
  • The introduction part of the manuscript should be revised with reference to the latest relevant literature. The reader can’t understand the storyline highlighting the problem under study, novelty and importance of this work.
  • Use “Phononic band gaps” instead of “Phonon band gaps” throughout the manuscript.
  • Please label different parts of Fig. 1. Also, it is not mentioned whether the figure depicts the cross-sectional view or the top view. For better understanding, I would suggest making a 3D model (not necessary to match the scale of the actual device) with proper labelling, SAW propagation direction and water flow direction.
  • The authors have considered the periodic array of microchannels as phononic crystals (PnCs). The band gap diagram (frequency versus reduced wave vector, k) is highly dependent on the lattice parameters of the phononic crystal. The band gap diagram along with the unit cell dimensions of the PnC is missing in the manuscript.
  • Which simulation software is used to achieve these results? Please mention the tool in the corresponding section.
  • For reproducibility of the results, briefly explain all the steps taken in the simulations.
  • Page 3, Line 99: Why different pressures have been applied for 100um and 1mm channel lengths? What will happen to the results if the same pressure is applied to the channels?
  • Graphs in Fig.3-8 are not reader-friendly. Please increase line size, x and y-axis data.
  • Mention the lower and upper edge of the band gap in Fig. 3(a) and Fig. 4. Does this band gap coincide with the band gap diagram asked from the authors in point 6 above?
  • Page 5, Line 121: What is the reason for considering only these values of sound velocities.
  • Page 5, Line 128: Please mention the percentage change for both cases in the same paragraph.
  • Page 5, Line 130: Use standard notation to describe 1o
  • Give reference to equations 5-8.
  • There is difference in magnitude of the local frequency response curves inside the bandgap of Fig, 3(a) and (b). For example, for k=3, the abs(p) is roughly 8x10-5 while in Fig. 3(b) abs(p) is 15x10-5. Please explain.
  • Page 6, Line 160: Please mention the liquid flow direction in Fig. 5(a). The colour bar is highly confusing with the same reading of 293.15 for all colours on the bar. Please mention the exact values for each colour.
  • Please explain Fig.5(b) in the corresponding paragraph.
  • The response curves in Fig.6 are not smooth like Fig. 3(a). I assume that the author has taken a larger step size in the frequency domain analysis. Please reduce the step size and put the updated results in Fig.6.
  • The authors have concluded that detection accuracy increases with length of channel but he has run simulations for just two channel lengths i.e. 100um and 1mm. I would rather suggest running simulations for an intermediate channel length like 500um and see the pattern in increasing detection accuracy for all three lengths.
  • The frequency shift in the local resonance peak inside the band gap determines the sensitivity of the sensor. There will be some arbitrary values of temperature/viscous dissipation for which the local peak inside the bandgap will shift towards the right and will ultimately touch the right edge of the band gap. So what should be the upper and lower limit of the sensitivity with varying temperature/viscous dissipation? I hope the author must address this point.
  • Why there exist periodic temperature curves with larger values at a particular arc length in Fig. 5(b) and Fig. 7? Do these curves correspond to the channel array?
  • Page 7, Line 185: remove “ ’ ” from properties’.
  • Page 8, Line 194: Space is missing in “maybe”.
  • Page 8, Line 200: Delete “This”.
  • Page 8, Line 200: A lengthy sentence “Therefore, to solve….”. Please divide into chunks.
  • Page 8, Line 203: Delete one dot after meantime.
  • Page 8, Line 209: Make correction in “phonon gap band”.
  • Page 8, Line 210: A lengthy sentence “ This article…..”. Please divide it into chunks.

Author Response

Thank you so much for your suggestion about revising. We have revised our article carefully according to your suggestion. Please see the attachment for point-by-point response. Many thanks again.

Reviewer 3 Report

The manuscript entitled “Simulating Study of Viscous Dissipation in Microchannel of Chemical Sensor Based on Phononic Crystal” by J. He et al. is clearly written, rigorous, original, and well-organized. The quality of figures is good and well-described. Therefore, it will be of great interest to an audience beyond the field of phononic crystals. In this manuscript, the authors attempt to tackle the issue related to the viscous dissipation due to existent temperature fluctuations in a system. Specifically, authors show that when the channel length reaches 1 mm, the frequency shift caused by viscous dissipation significantly influences their detection accuracy. It is also demonstrated that the viscous dissipation does influence the band gap of phononic crystal chemical sensor, something important for chemical sensors field of research.

Minor comments to address/incorporate:

The authors did a great job reviewing the literature. However, some fundamentally important works based on experimental evidence regarding high-frequency and short length scales at viscous-to-elastic crossovers are suggested to be mentioned:

  1. Bolmatov, M. Zhernenkov, D. Zav’yalov, S. Stoupin, Y. Q. Cai, A. Cunsolo, Revealing the mechanism of the viscous-to-elastic crossover in liquids, J. Phys. Chem. Lett. 6(15), 3048-3053 (2015).
  2. Bolmatov, M. Zhernenkov, D. Zav’yalov, S. Stoupin, A. Cunsolo, Y. Q. Cai, Thermally triggered phononic gaps in liquids at THz scale, Sci. Rep. 6, 19469 (2016).

This manuscript is subject to minor revisions.

Author Response

Thank you very much for your review. The articles you mentioned will be read carefully and cited in the next version of my manuscript. Many thanks again.

Round 2

Reviewer 1 Report

The manuscript was massively worked over, but to the effect that the handed in pdf for review was barely readable. Thus, I only focus on key points.

I looked up reference 8. There temperature changes of SU8 are described, however this reference does not describe how large the change in speed of sound due to a temperature shift by only 0.01K is. For water I estimated a speed of sound of 3.4m/s per 1K, i.e. the speed of sound changes due to 0.01K is 0.034m/s. I do really doubt that such a change is measurable. Also, the speed of sound changes 10m/s and 80m/s are misleading (figure 4). Figure5 presents changes in temperature in 1K steps, but this is still a factor 100 more than the heat dissipation calculated.

The paper should not be published unless the authors manage to present in a readable and understandable way that the thermal effects due to dissipation and the corresponding (!!!) acoustic effects have a chance to be measured. To do this all thermal effects have to be taken into account. Usually piezos get hot, when electrically driven. Most of the times these effects are negligible, however, if the maximum (!!!) temperature change is 0.01K also these minor effects must be taken into account.